# Recommender Transformers with Behavior Pathways

## Abstract

Sequential recommendation requires the recommender to capture the evolving behavior characteristics from logged user behavior data for accurate recommendations. However, user behavior sequences are viewed as a script with multiple ongoing threads intertwined. We find that only a small set of pivotal behaviors can be evolved into the user's future action. As a result, the future behavior of the user is hard to predict. We conclude this characteristic for sequential behaviors of each user as the *Behavior Pathway*. Different users have their unique behavior pathways. Among existing sequential models, transformers have shown great capacity in capturing global-dependent characteristics. However, these models mainly provide a dense distribution over all previous behaviors using the self-attention mechanism, making the final predictions overwhelmed by the trivial behaviors not adjusted to each user. In this paper, we build the *Recommender Transformer* (RETR) with a novel *Pathway Attention* mechanism. RETR can dynamically plan the behavior pathway specified for each user, and sparingly activate the network through this behavior pathway to effectively capture evolving patterns useful for recommendation. The key design is a learned binary route to prevent the behavior pathway from being overwhelmed by trivial behaviors. We empirically verify the effectiveness of RETR on seven real-world datasets and RETR yields state-of-the-art performance.

## 1 Introduction

Recommender systems [16, 24, 42] have been widely adopted in real-world industrial applications such as E-commerce and social media. Benefiting from the increase in computing power and model capacity, some recent efforts formulate recommendation as a time-series forecasting problem, known as *sequential recommendation* [18, 31, 6]. The core idea of this field is to infer upcoming actions based on user's historical behaviors, which are reorganized as time-ordered sequences. This intuitive modeling of recommendation is proved time-sensitive and context-aware to make precise predictions.

Recent advanced sequential recommendation models, such as SASRec [18], Bert4Rec [31] and SMRec [6], have achieved significant improvements. Transformers enable these models to recognize global-range sequential patterns, and to model how future behaviors are anchored in historical ones. The self-attention mechanism does make it possible to explore all previous behaviors of each user, with the whole neural network activated. However, misuse of user information, regardless of whether they are informative or not, floods models with trivial ones, makes models dense and inefficient, and results in key behaviors losing voice. And this clearly contradicts with the way our brain works.

The human being has many different parts of the brain specialized for various tasks, yet the brain only calls upon the relevant pieces for a given situation [40]. To some extent, user behavior sequences can be viewed as a script with multiple ongoing threads intertwined. And only key clues suggest what will happen next. In sequential recommendation, we find that only a small part of pivotal behaviors can be evolved into the user's future action. And we conclude this characteristics of sequential behaviors as the *Behavior Pathway*.

Submitted to 36th Conference on Neural Information Processing Systems (NeurIPS 2022). Do not distribute.

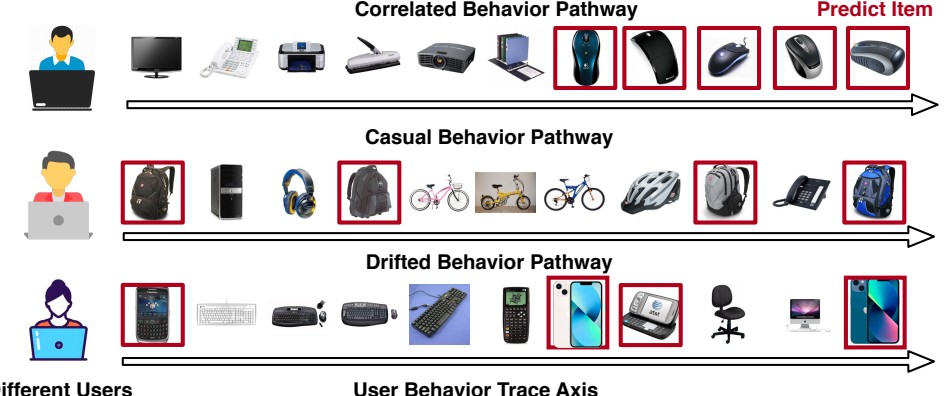

Figure 1: Three main characteristics of the behavior pathway for different users, making sequential recommendation extremely hard. The behavior pathway is outlined by the red boxes.

Different users have their unique behavior pathways and they can be grouped into three categories:

- **Correlated Behavior Pathway**: A user's behavior pathway is closely associated with behaviors at a certain period. As shown in the first line of Figure 1, the mouse is clicked many times recently, leading to the final decision to buy a mouse.
- **Casual Behavior Pathway**: A user's behavior pathway is interested in a specific item at casual times. As shown in the second line of Figure 1, the backpack is randomly clicked sequentially.
- **Drifted Behavior Pathway**: A user's behavior pathway in a particular brand might drift over time. As shown in the third line of Figure 1, the user was initially interested in a keyboard, but suddenly became interested in buying a phone at last.

It's challenging to capture these aspects dynamically for each user to make precise recommendations.

Motivated by the Pathways [8], a new way of thinking about AI, which builds a single model that is sparsely activated for all tasks with small pathways through the network called into action as needed, we propose a novel *Recommender Transformer* (RETR) with a *Pathway Attention* mechanism. RETR dynamically explores behavior pathways for different users and then captures evolving patterns through these pathways effectively. To be specific, the user-dependent pathway attention, which incorporates a pathway router, determines whether or not a behavior token will be maintained in the behavior pathway. Technically, the pathway router generates a customized binary route for each token based on their information redundancy. Recommender Transformers are stacked, and successive pathway routers constitute a hierarchical evolution pathway of user behaviors. To enable the pathway router modules to be end-to-end optimized, we adopt the Gumbel-Softmax [17] sampling strategy to overcome the non-differentiable problem of sampling from a Bernoulli distribution.

To effectively capture the evolving patterns via the behavior pathway, our pathway attention mechanism makes our RETR mainly attend to the obtained pathway. We force the model to focus on the most informative behaviors by using the query routed through the behavior pathway. We cut off the interaction from the off-pathway behaviors of the query. Compared with using all previous behaviors, our pathway attention mechanism is obviously more effective and can avoid the most informative tokens being overwhelmed by trivial behaviors. To validate the effectiveness of our approach, we conduct experiments on seven real-world competitive datasets for sequential recommendations and RETR achieves state-of-the-art performance. Our main contributions can be summarized as follows:

- Our work is the first to propose the concept of behavior pathway for sequential recommendation. We find the key to the recommender is to dynamically capture the behavior pathway for each user.
- We propose the novel recommender transformer (RETR) with a novel pathway attention mechanism, which can generate the behavior pathway hierarchically and capture the evolving patterns dynamically through the pathway.
- We validate the effectiveness of RETR on seven real-world datasets of different scales across different scenarios for sequential recommendations and achieve state-of-the-art performance.

## 2 Related Work

**Traditional recommendation approaches.** Capturing evolving behavior characteristics is crucial for many online applications, such as advertising, social media and E-commerce, and it is the key challenge for sequential recommendation [1, 18, 7, 39, 11, 26, 5, 45, 23]. Traditional recommendation approach, such as the collaborative filtering (CF) [15] based on matrix approximation [20, 21], always assumes that the user's behavior is static. However, in practice, user behaviors often change over time due to various reasons, making the CF deteriorate in a real-world application.

**Sequential recommendation approaches.** To overcome this challenge, some methods, such as FPMC [14] and HRM [34], use Markov chains to capture sequential patterns by learning user-specific transition matrices. Higher-order Markov Chains assume the next action is related to several previous actions. Benefit from this strong inductive bias, MC-based methods [14, 13] show superior performance in capturing short-term patterns. At the same time, there is a potential state space explosion problem when these approaches are faced with different possible sequences [35]. In recent years, many works have been using the deep neural network for sequential recommendation. The GRU4Rec [16] and the RepeatNet [27] adopt the recurrent network to capture dynamic patterns from the user behaviors dependent on sequence positions. The RNN-based models achieve competitive performance in capturing short-term behavior patterns but cannot capture long-term behavior patterns effectively. The CNN-based model, such as Caser [33], applies convolutional operations to extract transitions while tending to overlook the intrinsic relationship across user behaviors. The GNN-based methods, such as SRGNN [37], GCSAN [38], Jodie [22] and TGN [28] model behavior sequences as graph-structured data and incorporate an attention mechanism for a session-based recommendation. In addition, DIN [43] uses the gate mechanism to weight different user behaviors. However, concatenating all behaviors makes these models overlook the sequential characteristics.

**Attention-based models for Sequential Recommendation.** The attention-based models like SINE [32] have the strong capacity to capture behavior patterns via the attention mechanism, achieving state-of-the-art performance while involving many parameters. Especially, SASRec [18], BertRec [31], S3-Rec [44], TGSRec [10], LightSANs [9] and SSE-PT [36] introduce the transformer architecture into sequential recommendation, which might lead to the over-parameterized architecture of Transformer-based methods. These models capture the evolving patterns by the self-attention mechanism, interacting with all previous behaviors. However, dense interactions will make the model not adapt to different users and overwhelm behavior pathways. To tackle this challenge, our paper builds the Recommender Transformer (RETR) with a new Pathway Attention mechanism that is dynamically activated for the behavior pathway of all users. Distinct from the previous routing architecture like Switch Transformer [12] using the MoE [30] structure for natural language tasks, our RETR is designed explicitly for sequential recommendation. Our RETR uses the pathway router to adaptively route the sequential behavior of each user rather than routing the experts of feed-forward networks in switch transformer.

## 3 Method

Suppose that we have a set of users and items, denoted by $\mathcal{U}$ and $\mathcal{I}$ respectively. In the task of sequential recommendation, chronologically-ordered behaviors of a user $u \in \mathcal{U}$ could be represented by a user-interacted item sequence: $\{i_1, \cdots, i_n\}$. Formally, given a user $u$ with her or his behavior sequence $\{i_1, \cdots, i_n\}$, the goal of sequential recommendation is to predict the next item the user $u$ would interact with at the $(n+1)$-th step, denoted as $p(i_{n+1} \mid i_{1:n})$.

As aforementioned, we highlight the key to sequential recommendation as the exploration of user-tailored behavior pathways, through which evolving characteristics could be learned. Motivated by this, we propose a novel *Recommender Transformer* (RETR) with a new *Pathway Attention*, the core subassembly of which is a pathway router. Besides the modification of architecture, we additionally introduce a hierarchical update strategy for the behavior pathway in the feed-forward procedure.

### 3.1 Recommender Transformer

Considering the limitation of Transformers [4] for sequential recommendation, we renovate the vanilla architecture to the Recommender Transformer (Figure 2) with a Pathway Attention mechanism.

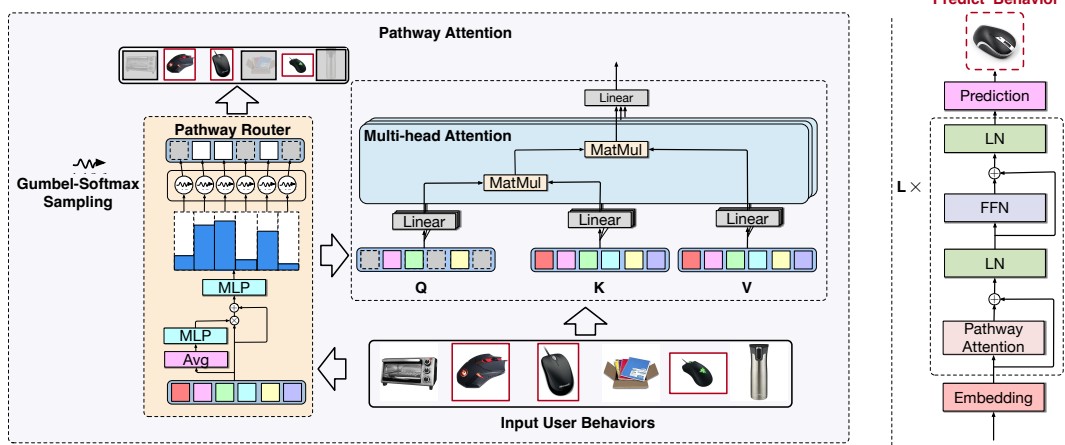

Figure 2: Recommender Transformer architecture (right). Pathway Attention (left) explores the behavior pathway by the pathway router (orange module) and captures the evolving sequential characteristics by the multi-head attention.

**Model inputs.** To obtain the model inputs, we follow the sliding window practice and transform the user's behavior sequence into a fixed-length-$N$ sequence $s = (s_1, s_2, \ldots, s_N)$. Then we produce an item embedding matrix $\mathcal{E}_{\mathcal{I}} \in \mathbb{R}^{|\mathcal{I}| \times d}$, where $d$ is the embedding dimensionality. We perform a look-up operation from $\mathcal{E}_{\mathcal{I}}$ to retrieve the input embedding matrix $\mathcal{E}_s \in \mathbb{R}^{N \times d}$ for sequence $s$. Besides, we also add a learnable position embedding $\mathcal{P}_s \in \mathbb{R}^{N \times d}$ for sequence $s$. Finally, we can generate the input embedding of each behavior sequence $s$ as $\mathcal{X}_s = \mathcal{E}_s + \mathcal{P}_s \in \mathbb{R}^{N \times d}$.

**Overall architecture.** Recommender Transformer is characterized by stacking the Pathway Attention blocks and feed-forward layers alternately, containing $L$ blocks. This stacking structure is conducive to learning behavior representations hierarchically. The overall equations of block $l$ are formalized as:

$$
\begin{aligned}
\widehat{\mathcal{Z}}^l, \mathcal{R}^l &= \text{Path-MSA}\left(\mathcal{Z}^{l-1}, \mathcal{R}^{l-1}\right) \\
\widehat{\mathcal{Z}}^l &= \text{LN}\left(\widehat{\mathcal{Z}}^l + \mathcal{Z}^{l-1}\right) \\
\mathcal{Z}^l &= \text{LN}\left(\text{FFN}\left(\widehat{\mathcal{Z}}^l\right) + \widehat{\mathcal{Z}}^l\right),
\end{aligned}
\tag{1}
$$

where $\mathcal{Z}^l \in \mathbb{R}^{N \times d}, l \in \{1, \cdots, L\}$ denotes the output of the $l$-th block. The initial input $\mathcal{Z}^0 = \mathcal{X}_s \in \mathbb{R}^{N \times d}$ represents the raw behavior embedding. $\mathcal{R}^{l-1}$ is the previous route from the block $l-1$ and we initialize all elements in the route $\mathcal{R}^0$ to 1. Path-MSA($\cdot$) is to conduct the pathway attention. LN($\cdot$) is to conduct layer normalization [3] and FFN represents the point-wise feed-forward network [4].

### 3.1.1 Pathway Attention

Note that the single-branch self-attention mechanism [4] in vanilla transformer cannot model the behavior pathway dynamically, resulting in key behaviors being overwhelmed by these non-pivotal ones. To solve this problem, we propose the Pathway Attention mechanism, as shown in Figure 2, which can dynamically attend to the behavior pathway of pivotal behavior tokens.

**Pathway router.** The pathway attention employs a sequence-adaptive pathway router to custom-tailor behavior pathway routes for users. The router generates a binary route $\mathcal{R}^l \in \{0, 1\}^N$ to determine whether a behavior token would be part of the behavior pathway or not. Each router takes the pre-order route $\mathcal{R}^{l-1}$ and user behavior tokens $\mathcal{Z}^{l-1} \in \mathbb{R}^{N \times d}$ of the block $l-1$ as its inputs. All elements in the route are initialized by 1 and are updated progressively in training.

Foremost, to suppress the potential disturbance to the model caused by the local drifted interest (Figure 1), it is crucial to incorporate the *global* information in the route generation. We apply the average pooling to all the preserved behavior tokens routed by $\mathcal{R}^{l-1}$, and produce the global sequential representation via a multilayer perceptrons (MLP) module. Then, we combine this global representation with the inputs and employ a residual connection to maintain the original input information. Finally, we feed them to another MLP layer to predict the probabilities of keeping or

dropping the behavior tokens. The above procedure can be formulated as follows:

$$\mathcal{Z}^l_{\text{emb}} = \mathcal{Z}^{l-1} + \mathcal{Z}^{l-1} \odot \text{MLP} \left( \frac{\sum_{i=1}^N \mathcal{R}^{l-1}_i \mathcal{Z}^{l-1}_i}{\sum_{i=1}^N \mathcal{R}^{l-1}_i} \right)$$

$$\boldsymbol{\pi} = \text{Softmax}\left(\text{MLP}(\mathcal{Z}^l_{\text{emb}})\right) \in \mathbb{R}^{N \times 2}, \tag{2}$$

where $\odot$ is the Hadamard product. For $t \in \{1, 2, \cdots, N\}$, we let $\boldsymbol{\pi}_t = [1 - \alpha_t, \alpha_t]$, where the logit $\alpha_t$ denotes the probability that the $t$-th behavior token is kept for the behavior pathway.

**Gumbel-Softmax sampling from $\boldsymbol{\pi}$ for router.**   Our goal is to generate the binary route from $\boldsymbol{\pi}$. However, sampling from $\boldsymbol{\pi}$ directly is non-differentiable, and it will impede the gradient-based training. Thus, we apply the Gumbel-Softmax [17] technique to such sample. Gumbel-Softmax is an effective way to approximate the original non-differentiable sample from a discrete distribution with a differentiable sample from a Gumbel-Softmax distribution. Instead of directly sampling a keep-or-drop decision $\widehat{\mathcal{R}}^l_t$ for the $t$-th behavior token from the distribution $\boldsymbol{\pi}_t$, we generate it as:

$$\widehat{\mathcal{R}}^l_t = \underset{j \in \{0,1\}}{\arg\max} \left( \log \pi_t(j) + G_t(j) \right), \tag{3}$$

where $G_t = -\log(-\log U_t)$ is a standard Gumbel distribution, and $U_t$ is sampled i.i.d. from a uniform distribution $\text{Uniform}(0, 1)$. To remove the non-differentiable $\arg\max$ operation in Eq 3, the Gumbel-Softmax uses the reparameterization trick [17] as a differentiable approximation to relax one-hot $\widehat{\mathcal{R}}^l_t \in \{0, 1\}$ to $v_t \in \mathbb{R}^2$:

$$v_t(j) = \frac{\exp((\log \pi_t(j) + G_t(j))/\tau)}{\sum_{i \in \{0,1\}} \exp((\log \pi_t(i) + G_t(i))/\tau)}, j \in \{0, 1\}, \tag{4}$$

where $\tau$ is the temperature parameter of the Softmax, which is commonly set to 1 in deep networks.

**Hierarchical update strategy for router.** The preliminary route $\widehat{\mathcal{R}}^l$, sampled from $\boldsymbol{\pi}$, is not a final decision. In our design, once a token fails to be routed in a certain block, it would permanently lose the privilege to be part of the behavior pathway in the following feed-forward procedure, constituting a hierarchical pathway router strategy. Thus finally we formulate the route $\mathcal{R}^l$ as the Hadamard product of $\widehat{\mathcal{R}}^l$ and the pre-order route $\mathcal{R}^{l-1}$ in the block $l - 1$:

$$\mathcal{R}^l = \widehat{\mathcal{R}}^l \odot \mathcal{R}^{l-1}. \tag{5}$$

**Multi-head pathway attention.**    The standard self-attention mechanism retrieves sequential characteristics exploiting all behavior tokens, making the behavior pathway overwhelmed by the trivial behaviors. In the new pathway attention, the pathway router would be firstly applied to the input behavior tokens to route information. The pathway router would not pare down the number of tokens, but only the interactions between the off-pathway and on-pathway tokens, as these off-pathway tokens may also convey contextual information.

Specifically, for the query, key, and value in the pathway attention: the query is routed by the pathway router, to prevent the pathway from being overwhelmed and to force the pathway attention to attend to the behavior pathway; the key and value are the original input behavior tokens, to ensure that the contextual information from off-pathway behavior tokens can be captured as well:

$$\mathcal{Q}_m, \mathcal{K}_m, \mathcal{V}_m = (\mathcal{Z}^{l-1} \odot \mathcal{R}^l) W^l_{\mathcal{Q}_m}, \mathcal{Z}^{l-1} W^l_{\mathcal{K}_m}, \mathcal{Z}^{l-1} W^l_{\mathcal{V}_m}$$

$$\widehat{\mathcal{Z}}^l_m = \text{Softmax}\left( \frac{\mathcal{Q}_m \mathcal{K}^{\text{T}}_m}{\sqrt{d/h}} \right) \mathcal{V}^l_m, \tag{6}$$

where $m \in \{1, 2, \cdots, h\}$ is the index of head in the multi-head self-attention; $W^l_{\mathcal{Q}_m}, W^l_{\mathcal{K}_m}, W^l_{\mathcal{V}_m} \in \mathbb{R}^{d \times d/h}$ are transformation matrices learned from data. Finally, the outputs $\left\{ \widehat{\mathcal{Z}}^l_m \in \mathbb{R}^{N \times d/h} \right\}_{1 \le m \le h}$ of multiple heads are concatenated into $\widehat{\mathcal{Z}}^l \in \mathbb{R}^{N \times d}$. We use $\widehat{\mathcal{Z}}^l, \mathcal{R}^l = \text{Path-MSA}\left(\mathcal{Z}^{l-1}, \mathcal{R}^{l-1}\right)$ to summarize the above pathway attention. Its output is further transformed by Eq. (1) to form the final output of the $l$-th block $\mathcal{Z}^l \in \mathbb{R}^{N \times d}$.

**Causality.**   In the prediction of the $(t + 1)$-th behavior, only the first $t$ observable behaviors should be taken into account. To avoid a future information leak and ensure causality, a look-ahead mask is employed and all links between $\mathcal{Q}_j$ and $\mathcal{K}_i$ $(j > i)$ are removed.

## 3.2 Prediction Layer and Training Objective

**Prediction layer.** In the final layer of our RETR, we calculate the user's preference score for the item $k$ in the step $(t+1)$ in the context of user behavior history as $p(i_{t+1} = k \mid i_{1:t}) = e_k \cdot \mathcal{Z}_t^L$, where $e_k$ is the representation of item $k$ from item embedding matrix $\mathcal{E}_{\mathcal{I}}$, and $\mathcal{Z}_t^L$ is the output of the $L$-th RETR blocks at step $t$ ($L$ is the number of RETR blocks).

**Training objective.** We adopt the pairwise ranking loss to optimize the model parameters as:

$$\mathcal{L} = -\sum_{u \in \mathcal{U}} \sum_{t=1}^{n} \log \sigma(p(i_{t+1} \mid i_{1:t}) - p(i_{t+1}^- \mid i_{1:t})), \tag{7}$$

where we pair each ground-truth item $i_{t+1}$ with a randomly sampled negative item $i_{t+1}^-$. In each epoch, we randomly generate one negative item for each time step in each sequence. This pairwise ranking loss is widely adopted in previous literature of sequential recommendation [18, 45].

## 4 Experiments

We extensively evaluate the proposed Recommender Transformer on seven real-world benchmarks.

**Datasets.** Here are descriptions of the seven datasets: (1) **Netflix**: Netflix dataset is a large-scale movie rating dataset released by Netflix. (2) **MSD**: The Million Song Dataset (MSD) is a large-scale, metadata-rich and open-source dataset on Kaggle. (3) **Taobao**: Taobao dataset [32] contains user behaviors in Taobao's recommender system. In experiments, we only use the click behaviors. (4) **Yelp** [2]: Yelp is a dataset for business recommendation. We only use the transaction records after January 1st, 2019. (5) **Tmall**: Tmall contains users' shopping logs on

Table 1: Statistics of the datasets.

| Dataset | Users | Items | Actions |
|---|---|---|---|
| Netflix | 463,435 | 17,769 | 57,000,000 |
| MSD | 571,355 | 41,140 | 34,000,000 |
| Taobao | 987,994 | 4,162,024 | 100,150,807 |
| Yelp | 30,431 | 20,033 | 316,354 |
| MovieLens-1M | 6,040 | 3,416 | 1,000,000 |
| Tmall | 66,909 | 37,367 | 427,797 |
| Steam | 334,730 | 13,047 | 3,700,000 |

Tmall online shopping platform, which is from the IJCAI-15 competition. (6) **Steam** [18]: Steam dataset is collected from a large online video game distribution platform. This dataset includes 2,567,538 users, 15,474 games and 7,793,069 English reviews from October 2010 to January 2018. (7) **MovieLens**: this is a widely used benchmark dataset for evaluating collaborative filtering algorithms. The version we use is MovieLens-1M, which includes 1 million user ratings. The statistics of the seven datasets are summarized in Table 1. All datasets are widely used in the recommendation task. It is notable that Netflix, MSD, Taobao and Steam are large-scale recommendation datasets.

We group the interaction records by users or sessions for all datasets and sort them by the timestamps in ascending order. We follow the operation in SASRec [18] and split the historical sequence for each user into three parts: (1) the most recent behavior for testing, (2) the second most recent behavior for validation, and (3) all remaining behaviors for training. During testing, the input sequences contain training behaviors and validation behaviors. We filter less popular items and inactive users with fewer than five interaction records.

**Evaluation metrics.** Following the previous literature [41, 45, 18], we apply top-k Hit Ratio (HR@k), top-k Normalized Discounted Cumulative Gain (NDCG@k) and Mean Reciprocal Rank (MRR) for evaluation. We report HR@10, NDCG@10 and MRR of the results. Besides, following the standard strategy in SASRec [18], we pair the ground-truth item with 100 randomly sampled negative items that the user has not interacted with. All metrics are calculated according to the ranking of the items and we report the average score.

**Baseline methods.** We compare our RETR with GRU4Rec [16], a simple baseline that applies GRU to model item sequences, and state-of-the-art sequential recommendation models: SASRec [18], BertRec [31], SMRec [6]. S3-Rec [44], SINE [32], TGSRec [10] and LightSANs [9]. These methods adopt the attention mechanism to make precise recommendations. Besides, we also compare our RETR with state-of-the-art graph-based sequential recommendation methods: Jodie [22] and TGN [28]. All baseline methods are configured using default parameters of the original paper or optimal parameters which can produce their best results through a grid search.

244 **Implementation details.** Our model is supervised by the pairwise rank loss in Equ 7, using the
245 ADAM [19] optimizer with an initial learning rate of 0.001. Batch size is set to 512. The maximum
246 number of training epochs for all methods is set to 200. All hyperparameters are tuned on the
247 validation set. The training process is early stopped within 10 epochs. Our RETR has $L = 2$ layers,
248 and each layer has $h = 4$ heads (the ablation study of multi-head attention can be found in the
249 appendix A) and $d$ is set to be 256. The maximum sequence length $N$ is set to 200 for MovieLens-1m
250 and 100 for the other six datasets. All experiments are repeated three times, implemented in PyTorch
251 [25], and conducted on a single NVIDIA 3090 GPU.

Table 2: Performance comparison to state-of-the-art models: GRU4Rec [16], BERTRec [31], SASRec
[18], SMRec [6], S3-Rec [44], SINE [32], TGSRec [10], LightSANs [9], Jodie [22], TGN [28].

| Datasets | Meric | GRU4Rec | BERT4Rec | SASRec | SMRec | S3-Rec | SINE | TGSRec | LightSAN | Jodie | TGN | RETR |
|---|---|---|---|---|---|---|---|---|---|---|---|---|
| Netflix | HR@10 | 0.4358 | 0.4792 | 0.4622 | 0.4848 | 0.4917 | 0.4902 | 0.4887 | 0.4852 | 0.4813 | 0.4802 | **0.5142** |
| | NDCG@10 | 0.2912 | 0.3330 | 0.3202 | 0.3492 | 0.3571 | 0.3601 | 0.3512 | 0.3441 | 0.3368 | 0.3318 | **0.3725** |
| | MRR | 0.2431 | 0.2652 | 0.2519 | 0.2725 | 0.2819 | 0.2796 | 0.2778 | 0.2785 | 0.2687 | 0.2612 | **0.3134** |
| MSD | HR@10 | 0.3546 | 0.4819 | 0.4766 | 0.5083 | 0.5315 | 0.5264 | 0.5137 | 0.4994 | 0.4825 | 0.4782 | **0.5912** |
| | NDCG@10 | 0.3772 | 0.4891 | 0.4831 | 0.5112 | 0.5381 | 0.5304 | 0.5279 | 0.5163 | 0.4872 | 0.4832 | **0.5981** |
| | MRR | 0.2503 | 0.3120 | 0.3079 | 0.3302 | 0.3494 | 0.3667 | 0.3612 | 0.3451 | 0.3224 | 0.3102 | **0.3901** |
| Taobao | HR@10 | 0.0788 | 0.1261 | 0.1182 | 0.1272 | 0.1336 | 0.1580 | 0.1537 | 0.1590 | 0.1447 | 0.1421 | **0.1768** |
| | NDCG@10 | 0.0182 | 0.0425 | 0.0391 | 0.0631 | 0.0827 | 0.0873 | 0.0745 | 0.0694 | 0.0582 | 0.0571 | **0.1195** |
| | MRR | 0.0273 | 0.0489 | 0.0436 | 0.0721 | 0.0919 | 0.0934 | 0.0802 | 0.0741 | 0.0628 | 0.0603 | **0.1117** |
| Yelp | HR@10 | 0.7265 | 0.7597 | 0.7373 | 0.7548 | 0.7597 | 0.7564 | 0.7533 | 0.7552 | 0.7492 | 0.7473 | **0.7730** |
| | NDCG@10 | 0.4375 | 0.4778 | 0.4642 | 0.4789 | 0.4937 | 0.4902 | 0.4887 | 0.4863 | 0.4792 | 0.4784 | **0.5136** |
| | MRR | 0.3630 | 0.4026 | 0.3927 | 0.4023 | 0.4107 | 0.4093 | 0.4072 | 0.4086 | 0.3997 | 0.3985 | **0.4354** |
| MovieLens | HR@10 | 0.5581 | 0.8269 | 0.8233 | 0.8302 | 0.8352 | 0.8311 | 0.8303 | 0.8294 | 0.8277 | 0.8259 | **0.8467** |
| | NDCG@10 | 0.3381 | 0.5965 | 0.5936 | 0.6079 | 0.6172 | 0.6134 | 0.6081 | 0.6119 | 0.6009 | 0.5998 | **0.6351** |
| | MRR | 0.3002 | 0.5614 | 0.5573 | 0.5703 | 0.5812 | 0.5801 | 0.5734 | 0.5791 | 0.5651 | 0.5627 | **0.5921** |
| Tmall | HR@10 | 0.6432 | 0.6196 | 0.6275 | 0.6476 | 0.6687 | 0.6512 | 0.6506 | 0.6399 | 0.6384 | 0.6362 | **0.7138** |
| | NDCG@10 | 0.5169 | 0.5025 | 0.5049 | 0.5192 | 0.5423 | 0.5411 | 0.5372 | 0.5415 | 0.5307 | 0.5198 | **0.6103** |
| | MRR | 0.4975 | 0.4026 | 0.4804 | 0.4934 | 0.5194 | 0.5147 | 0.5121 | 0.5119 | 0.5003 | 0.4997 | **0.5822** |
| Steam | HR@10 | 0.4190 | 0.8656 | 0.8729 | 0.8792 | 0.8813 | 0.8765 | 0.8773 | 0.8832 | 0.8780 | 0.8731 | **0.9001** |
| | NDCG@10 | 0.2691 | 0.6283 | 0.6306 | 0.6408 | 0.6573 | 0.6502 | 0.6491 | 0.6519 | 0.6451 | 0.6399 | **0.6795** |
| | MRR | 0.2402 | 0.5883 | 0.5925 | 0.6011 | 0.6135 | 0.5972 | 0.6003 | 0.6104 | 0.5873 | 0.5798 | **0.6326** |

## 4.1 Main Results

253 The results of different methods on seven datasets are shown in Table 2. We can easily find that
254 attention-based models, SASRec [18], BertRec [31], SMRec [6], S3-Rec [44], SINE [32], TGSRec
255 [10] and LightSANs [9], achieve better performance than RNN-based model GRU4Rec [16] on most
256 datasets. It indicates that the attention mechanism is crucial for sequential recommendation, making
257 the model have a better capacity to capture sequential characteristics. These models can capture the
258 interaction information between all previous user behaviors via the attention mechanism. Besides,
259 the graph-based models like Jodie [22] and TGN [28] also achieve competitive performance. Besides,
260 our RETR can achieve state-of-the-art performance by a large margin on most datasets compared
261 with all baseline models.

262 **Results on Yelp, MovieLens1M and Tmall.** Specifically, our RETR achieves competitive perfor-
263 mance on the Yelp and Tmall. These datasets are sparse, containing less action information. Thus
264 they have lots of noisy logged information. By effectively capturing the behavior pathway, our
265 RETR is not affected by this trivial behavior information and captures the most informative behavior
266 representation to achieve better performance. Note that under the Tmall benchmark, RETR gains **7%**
267 HR@10, **12%** NDCG@10 and **14%** MRR against the strongest baseline SMRec [6]. Besides, for
268 the MoveLens1M, our RETR also achieves the best performance among all competing baselines.

269 **Results on large-scale datasets.** Our RETR can consistently achieve state-of-the-art results on
270 large-scale datasets (Netflix, MSD, Taobao, and Steam). These datasets are challenging and difficult
271 to capture pivotal behavior pathway useful for precise recommendation from the rich but noisy user's
272 behaviors. Especially for the Taobao dataset, our RETR gains relative improvements of **12%** HR@10,
273 **37%** NDCG@10 and **20%** MRR against the strongest baseline SINE [32]. It provides evidence that
274 our RETR can achieve competitive performance in both small- and large-scale datasets.

275 The substantial performance gains of our RETR indicate that focusing more on the behavior pathway
276 enables RETR to capture sequential characteristics more efficiently and effectively than the vanilla
277 self-attention mechanism, which considers all previous user behaviors.

## 4.2 Ablation Study

**Effectiveness of each model component.** In the left column of Table 3, we analyze the efficacy of each component in RETR on the Yelp dataset and have the following observations. First, we remove the pathway router module and randomly choose whether it can be maintained or dropped for each input behavior token. Removing the pathway router decreases the prediction performance a lot (MRR: $0.4354 \rightarrow 0.3887$), showing the necessity of learning behavior pathway effectively based on a data-dependent module. Second, discarding the hierarchical update strategy for the behavior pathway also decreases the prediction performance, suggesting that this strategy is crucial for RETR to get a more accurate behavior pathway.

**Number of blocks.** In the right column of Table 3, we adjust the number of blocks for RETR on Yelp. We find that the performance first increases rapidly with the growth of the block number and achieves the best performance at $L = 2$. We perform a similar grid search on other datasets.

Table 3: Ablation study of (**Left**) the effectiveness of each model component and (**Right**) the number of blocks for each RETR block. Experiments are conducted on the Yelp Dataset.

| Model | MRR | Model (# number of blocks) | MRR |
|---|---|---|---|
| **RETR** | **0.4354** | RETR ($L = 1$) | 0.4197 |
| RETR w/o Pathway Router | 0.3887 | RETR ($L = 2$) | **0.4354** |
| RETR w/o hierarchical update | 0.4234 | RETR ($L = 3$) | 0.4342 |
| SASRec | 0.3927 | **RETR** ($L = 4$) | 0.4340 |

Table 4: Ablation study of (**Left**) the effectiveness of different temperatures; Comparison Parameters and GFLOPs (**Right**). All ablation study experiments are conducted on the Yelp Dataset.

| Model (temperature) | MRR | Model | Parameters (M) | GFLOPs | MRR |
|---|---|---|---|---|---|
| RETR ($\tau = 0.4$) | 0.4312 | RETR | 5.021 | 9.558 | **0.4354** |
| RETR ($\tau = 0.8$) | **0.4354** | SASRec [18] | 4.916 | 9.552 | 0.3927 |
| RETR ($\tau = 1$) | 0.4292 | SINE [32] | 5.112 | 9.741 | 0.4011 |
| RETR ($\tau = 2$) | 0.4183 | SMRec [6] | 5.173 | 9.864 | 0.4023 |

**Effectiveness of temperature.** In the left column of Table 4, we analyze the efficacy of different temperatures for Gumbel-Softmax sampling in RETR on the Yelp dataset. We observe that the performance first increases rapidly with the growth of the temperature and achieves the best performance when $\tau = 0.8$, while the performance degenerates a lot when $\tau > 1$. The temperature $\tau$ softens the softmax with $\tau > 1$. However, when $\tau \rightarrow \infty$, the Gumbel-Softmax distribution $p_\tau(y_t) \rightarrow 0.5$ becomes more smooth, leading to the maximum uncertainty. To make the sampling results more convincing, we apply the temperature calibration $\tau < 1$ during training to avoid overconfident predictions. These results show that Gumble-Softmax sampling with lower temperature ($\tau < 1$) avoids overconfident predictions, leading to better performance.

**Evaluation on efficiency.** The efficiency is compared between SASRec [18], SINE [32] and SMRec [6] on the Yelp dataset. The computation cost is measured with gigabit floating-point operations (GFLOPs) on the self-attention module with position encoding. Meanwhile, the model scale measured with parameters is also presented. As shown in Table 4, our RETR has almost the same number of parameters or GFLOPs, compared with SASRec, indicating that our pathway router is a light-weight module. Our pathway attention does not bring more costs. It's worth noticing that the parameter scales and GLOPs of other competing transformers (apart from SASRec) are larger than RETR, but our RETR achieves higher performance. This result shows that our RETR is more efficient and effective than other competing attention-based models.

## 4.3 Case study

**Setups.** We also provide qualitative visualizations for our RETR, and SASRec [18]. Technically, we use the GradCAM [29] to generate behavior heatmaps of the output of the last layer in each model. Three random examples of users' historical behaviors in the Steam dataset are shown in sequential order through subplots (a)–(c) in Figure 3. We provide attention heatmaps of each example at the last

ten time steps. We can observe three main behavior pathway characteristics corresponding to three behavior sequences respectively: (a) *Casual behavior pathway:* RPG games are randomly clicked by the user, while the user has a continuing interest in RPGs. (b) *Correlated behavior pathway:* The user has recently been interested in indie games. (c) *Drifted behavior pathway:* The user has recently been interested in simulation games but chooses an indie game at last.

**Visualization results.** We elaborate the three representative categories of behavior pathway in recommender systems with model-learned attention heatmaps. (1) *Casual behavior pathway:* As shown in Figure 3(a), the RGB game is randomly clicked at casual times. Our RETR can capture all the RPG casual behavior pathways, while the SASRec focuses on the incorrect recent adventure games. The SASRec cannot capture the early clicked RPG game. This phenomenon proves that our RETR can deal with the casual behavior pathway effectively. (2) *Correlated behavior pathway:* For the correlated behavior pathway, we also provide an example which is shown in Figure 3(b). The indie game is clicked many times recently, leading to the final decision to an indie game. Our RETR can effectively capture the correlated behavior pathway. However, the SASRec provides higher attention scores on the recent RPG games. On the contrary, our RETR pays no attention to these wrong results, showing that it has a greater ability to cope with the correlated behavior pathway. (3) *Drifted behavior pathway:* As shown in Fig 3(c). The user was initially interested in the indie game, but suddenly became interested in simulation games recently and chose an indie game at last. Our RETR captures the drifted behavior pathway for the indie game and has not concentrated on the old drifted pathway – simulation games, while the SASRec is affected by the trivial behaviors of simulation games. These visualization results strongly show that our RETR can capture various behavior pathways dynamically for each user.

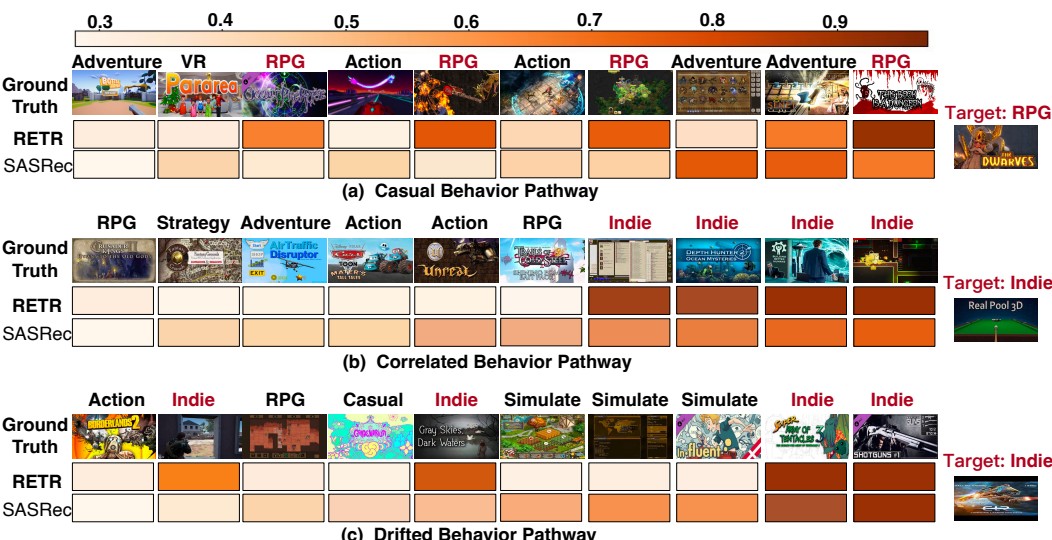

Figure 3: Visualizations of behavior heatmaps for RETR and SASRec of three random users in Steam dataset. They are corresponding to casual, correlated and drifted behavior pathways respectively.

## 5 Conclusion

A sequential recommender is designed to make accurate recommendations based on users' historical behaviors. The sequential recommendation system has benefited many practical applications such as online advertising. However, the users' behaviors are dynamic and come in a continually evolving manner. A user's current decision may only call upon the interest from the certain relevant behaviors of the past. We conclude these sequential characteristics as the behavior pathway. Previous models cannot capture the behavior pathway dynamically. We propose the Recommender Transformer (RETR) with a novel pathway attention mechanism to tackle these challenges. The pathway attention develops a pathway router to dynamically get the behavior pathway for each user and capture the evolving patterns. Our RETR achieves state-of-the-art performance on seven real-world datasets for sequential recommendation. The visualization results also show that our RETR can dynamically capture the behavior pathway for each user.

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
