

Figure 1: Illustrations of how RETR, SASRec and FMLP-Rec [8] differs on utilizing the historical behaviors of a random user in Steam Dataset. We provide the visualization of behavior heatmaps for RETR, SASRec and FMLP-Rec of a random user in Steam dataset.

## A  Further Ablation Study

Table 1: Ablation study of the head number $h$ for RETR on the Yelp Dataset.

| Model (# $h$) | MRR |
|---|---|
| RETR ($h = 1$) | 0.4310 |
| RETR ($h = 2$) | 0.4336 |
| **RETR** ($h = 4$) | **0.4354** |
| RETR ($h = 8$) | 0.4351 |

**Number of heads.**  In Table 2, we adjust the number of heads for RETR on Yelp. We find that the performance first increases rapidly with the growth of the head number and achieves the best performance at $h = 4$. We perform a similar grid search on other datasets.

Table 2: Ablation study of the maximum sequence length for RETR (**Left**) and SASRec (**Right**). Experiments are conducted on the Beauty Dataset.

| Model (# maximum sequence length ) | MRR | Model (# maximum sequence length ) | MRR |
|---|---|---|---|
| RETR ($N = 25$) | 0.2910 | SASRec ($N = 25$) | 0.2817 |
| RETR ($N = 50$) | 0.2979 | **SASRec** ($N = 50$) | **0.2852** |
| **RETR** ($N = 100$) | **0.3067** | SASRec ($N = 100$) | 0.2848 |
| RETR ($N = 200$) | 0.3058 | SASRec ($N = 200$) | 0.2841 |

**Maximum sequence length.**  In the Table 2, we adjust the maximum sequence length $N$ for RETR and SASRec on Beauty. As shown in left column of Table 2, we find that the performance of our RETR first increases rapidly with the growth of the block number and achieves the best performance at $N = 100$. We perform a similar grid search on other datasets. On the contrary, as shown in right column of Table 2, we find that the performance of SASRec achieves the best performance at $N = 50$ and then drops when $N > 50$. When $N = 100$, our RETR yields better performance compared with $N = 50$. However, the SASRec degenerates when $N = 100$. It indicates that our RETR can further exploit more useful information from the **longer sequence**.

## B  Results on the Beauty, Sports and Toys Dataset

Beauty, Sports, and Toys are three subcategories obtained from Amazon review [4] datasets. Our RETR can achieve state-of-the-art performance by a large margin on most datasets compared with all baseline models on these three datasets. It is noticed that the Beauty, Sports, and Toys are sparse, containing less action information. Thus they have lots of noisy logged information. By effectively capturing the behavior pathway, our RETR is not affected by this trivial behavior information and captures the most informative behavior representation to achieve better performance.

Table 3: Performance comparison of the baselines (PopRec, Caser [6], GRU4Rec [2], BERTRec [5], SASRec [3], SASRec+ [7] and SMRec [1]) and our method on the Beauty, Sports, and Toys. We use HR@10, NDCG@10 and MRR as our metrics. For these three metrics, a higher value indicates a better performance.

| Datasets | Meric | PopRec | Caser | GRU4Rec | BERT4Rec | SASRec | SASRec+ | SMRec | RETR |
|----------|-------|--------|-------|---------|----------|--------|---------|-------|------|
| Beauty | HR@10 | 0.3386 | 0.3942 | 0.4106 | 0.4739 | 0.4696 | 0.4798 | 0.4826 | **0.5034** |
| | NDCG@10 | 0.1803 | 0.2512 | 0.2584 | 0.2975 | 0.3156 | 0.3261 | 0.3238 | **0.3425** |
| | MRR | 0.1558 | 0.2263 | 0.2308 | 0.2614 | 0.2852 | 0.2901 | 0.2918 | **0.3067** |
| Sports | HR@10 | 0.3423 | 0.4014 | 0.4299 | 0.4722 | 0.4622 | 0.4776 | 0.4853 | **0.5083** |
| | NDCG@10 | 0.1902 | 0.2390 | 0.2527 | 0.2775 | 0.2869 | 0.2987 | 0.3061 | **0.3175** |
| | MRR | 0.1660 | 0.2100 | 0.2191 | 0.2378 | 0.2520 | 0.2635 | 0.2665 | **0.2768** |
| Toys | HR@10 | 0.3008 | 0.3540 | 0.3896 | 0.4493 | 0.4663 | 0.4729 | 0.4754 | **0.5104** |
| | NDCG@10 | 0.1618 | 0.2183 | 0.2274 | 0.2698 | 0.3136 | 0.3183 | 0.3198 | **0.3395** |
| | MRR | 0.1430 | 0.1967 | 0.1973 | 0.2338 | 0.2842 | 0.2912 | 0.2910 | **0.3048** |

## Broader Impact

**Real-world applications.** This paper studies sequential recommendation from a brand new perspective. The social impact of the proposed approach depends on its specific application, as is the case in recommendation tasks. Our method achieves consistent state-of-the-art performance in seven real-world applications. Thus, our work can be valuable for the community. Overall, we are positive about the potential ethical issues of our approach.

**Model robustness.** Based on the extensive experiments, we do not find exceptional failure cases. However, if the behaviors are extremely casual, RETR and any other models may degenerate.

Our work only focuses on the scientific problem, so there is no potential ethical risk.