# OpenReview forum: "Recommender Transformers with Behavior Pathways"
_NeurIPS.cc/2022/Conference — NeurIPS 2022 Submitted_

### Official Review · Reviewer_zis8 · 2022-07-05

**Rating:** 5
**Confidence:** 3
**Soundness:** 2 fair
**Presentation:** 2 fair
**Contribution:** 2 fair

**Summary:**

The authors propose Recommender Transformer (RETR) with a Pathway Attention mechanism which can generate the behavior pathway hierarchically and capture the evolving patterns dynamically through the pathway. The key design is a learned binary route to prevent the behavior pathway from being overwhelmed by trivial behaviors. The authors also show RETR has high accuracy and efficiency compared with other self-attention or transformer based sequential recommendation methods through experiments.

**Questions:**

- What’s the essential difference between RETR and sequential model with attention mechanism when putting aside the concept of pathway?
- In the introduction, the authors list three kinds of behavior pathways, so how can RETR capture them to make precise recommendations in each case?
- In Line 172, what’s the meaning of  “lose the privilege to be part of the behavior pathway”? Since the off-pathway tokens are also considered, it seems that all items will always be considered whether they are in the pathway or not.


**Limitations:**

The authors have adequately addressed the limitations and potential negative societal impact of their work

**Strengths And Weaknesses:**

Pros:
1. The paper is generally easy to follow.
2. The idea of using pathway in recommendation algorithms seems to be new.
3. The authors conducted extensive experiments on seven datasets to prove RETR can make accurate recommendations.

Cons:
1. The essential difference between RETR and other sequential recommendation methods with attention mechanism is not clear. It seems to me that RETR not only utilizes the “on-pathway tokens” but also leverages the “off-pathway tokens” as they “also convey contextual information” and the difference between the two kinds of tokens is their weight. However, in SASRec and all the other self-attention-based methods, different tokens already have varying attention weights, so that more important historical items may have higher attention weights and less important historical items may have lower attention weights. I don't understand why is it necessary to design a pathway.
2. The targeted problem is not new, and several recent works have been proposed to address the same issue of self-attention. Apart from reference [6] in the paper, there are several others trying to improve self-attention-based recommendation methods. For example, the LightSANs work published in SIGIR ‘21. It would be useful to also conduct experimental comparisons with these more recent baselines in addition to [6]. Besides, there are many sparse attention works in the literature and it is interesting to replace the proposed pathway-based method with other sparse attention methods to see if pathway-based method is superior.
3. Besides self-attention-based methods, other types of sequential recommendation methods, e.g., temporal graph-based sequential recommendation methods, have also achieved state-of-the-art results. Considering that attention weights can be regarded as edge weight in a graph, it might be useful to compare with some recent temporal graph based sequential recommendation methods, e.g., Jodie in KDD 19 and TGN in ICML 20. Especially, TGN also adopted attention in its model.
4. There is no evidence/experiment to show that RETR will not be overwhelmed by trivial behaviors. As this is the main claim of the paper, it should be necessary to have more justifications. The case study in Section 4.3 seems to be a cherry-pick result.
5. All the datasets that used in this paper are relatively small. Larger datasets such as MSD and Netflix should be more desirable.

Minor issues:
1. “in the second line” in Line 41 and “in the first line” in Line 44 should be exchanged.
2. In column “Actions” of Table 1, the commas are not consistent.
3. The authors should use “GRURec” or “GRU4Rec” consistently in the paper to avoid misunderstandings.
4. The ablation study of multi-head attention is missing.

---

> ### Author Response · Authors · 2022-08-02
> **Our Response to Reviewer zis8**
>
> **Q1:** What’s the essential difference between RETR and sequential model with attention mechanism when putting aside the concept of pathway?
>
> Our RETR has two essential differences compared with sequential model with attention mechanism:
>
> - Our RETR designs the pathway router to capture the behavior pathway, while the other sequential models have not considered it before.
>
>   - As shown in $\underline{\textrm{Figure 3}}$ of the revised paper, the previous self-attention mechanism mainly focuses on the recent behaviors, and cannot capture the accurate behavior pathway. The detailed analysis can be seen in **Q2**. Only our RETR can capture the precise behavior pathway.
>
> - The pathway attention for RETR is the cross-attention between the pathway behavior tokens and off-pathway tokens. Our pathway cross-attention mechanism can avoid the trivial interaction between the off-pathway tokens.
>
>   - As described in $\underline{\textrm{Line 183-188}}$, we routed the query using the captured pathway, which masks the off-pathway tokens as 0. Thus, the query for the pathway attention only contains information from the behavior pathway.
>
>   - This cross-attention mechanism forces the pathway attention to attend to the behavior pathway; To ensure that the contextual information from off-pathway behavior tokens can be captured, the key and value of the cross-attention are the original input behavior tokens.
>
>   - Our pathway cross-attention mechanism avoids the trivial interaction between the off-pathway tokens. However, the previous attention mechanism for sequential models is self-attention, which will be overwhelmed by the trivial information in the off-pathway behavior tokens.
>
> We further conduct evaluation experiments on Tmall. Firstly, we train RETR on Tmall. Secondly, we use the trained RETR on Tmall to capture the behavior pathway for each user in Tmall. Finally, We use the pathway behaviors and off-pathway behaviors as the inputs to train SASRec respectively.
>
> Method  | NDCG@10 ($\uparrow$) | HR@10 ($\uparrow$) | MRR ($\uparrow$)
> ---- | --- | --- | ---
> SASRec | 0.5049       |  0.6275          |  0.4804
> SASRec w/ pathway inputs |    0.5778    |  0.6812         |   0.5425
> SASRec w/ off-pathway inputs |    0.4383    |  0.5697         |   0.4215
> RETR | 0.6103 | 0.7138 | 0.5822
>
> From the above results, we can see that SASRec achieves better performance using the behavior pathway as the inputs compared with the original SASRec using the whole user's behavior as the inputs. On the contrary, the off-pathway inputs hurt SASRec's performance seriously.
>
> Finally, our RETR achieves the best performance, indicating that the pathway-offpathway cross-attention is more effective than the pathway self-attention.
>
>
>
> **Q2:** In the introduction, the authors list three kinds of behavior pathways, so how can RETR capture them to make precise recommendations in each case?
>
> The pathway router for RETR is to detect the accurate behavior pathway. As described in $\underline{\textrm{Line 152-153}}$, the pathway router embeds global information from the whole behavior sequence.
>
> Capturing these three kinds of behavior pathways has diverse challenges. For the **casual behavior** pathway, the recommender needs to capture the global interest of the whole sequence to avoid forgetting the early interests; For the **correlated behavior** pathway, even though previous methods can focus on the recent behaviors, these models also take the off-pathway tokens into consideration; For the **drifted behavior** pathway, the recommender needs to make decisions in a global view without focusing more on the old drifted pathway.
>
> To overcome these challenges, the pathway router embeds global information from the whole behavior sequence and maintains the original information from the input representation. It can make the router capture the global trend of the user's behaviors and remember the recent behavior information.
>
> We have provided additional visualization results for casual, correlated and drifted behavior pathways respectively in $\underline{\textrm{Figure 3 of the revised paper}}$. These three random samples from the steam dataset provide strong evidence that our RETR can capture various pathways.

---

> > ### Author Response · Authors · 2022-08-02
> > **Our Response to Reviewer zis8**
> >
> > **Q3:** In Line 172, what’s the meaning of “lose the privilege to be part of the behavior pathway”? Since the off-pathway tokens are also considered, it seems that all items will always be considered whether they are in the pathway or not.
> >
> > As described in $\underline{\textrm{Line 172-176}}$ of the revised paper, the behavior pathway is updated hierarchically in the subsequent feed-forward procedure. Once the behavior token fails to be routed as part of the pathway in a certain block, this token cannot be part of the final behavior pathway. This is the meaning of "losing the privilege to be part of the behavior pathway". We further clarify this in the revised paper.
> >
> > Our RETR does not directly consider all items. As described in **Q1**, our pathway attention is cross-attention between the pathway and off-pathway behavior tokens. The query for pathway attention only contains the pathway information, while the key and value are the whole input tokens. This cross-attention is proved by ablations to be the most effective way to make precise recommendation.
> >
> >
> > **Q4:** Why is it necessary to design a pathway.
> >
> > As described in **Q1**, we show that SASRec using the pathway as the inputs can achieve better performance compared with the original SASRec using the whole behaviors as the inputs. It indicates that the previous self-attenton mechanism makes the pathway overwhelmed by the other trivial behaviors.
> >
> > Thus, we design a router to capture the accurate pathway and propose a novel pathway attention, which is a cross-attention mechanism between the pathway and off-pathway tokens. This cross-attention can make our RETR not overwhelmed by the trivial off-pathway behaviors.
> >
> >
> > **Q5:** Compare with recent baselines.
> >
> > We add comparisons with the recent sequential recommendation models S3-Rec, SINE, TGSRec and LightSANs on all benchmarks. The full table is included the revised paper.
> >
> > We here give snapshot results on Tmall (full comparisons can be found in $\underline{\textrm{Table 2}}$ in the revision). From these results, we can see that our RETR remarkably outperforms the SOTA methods.
> >
> >
> > Method  | NDCG@10 ($\uparrow$)  | HR@10 ($\uparrow$) | MRR ($\uparrow$)
> > ---- | --- | --- | ---
> > S3-Rec [1] | 0.5423        | 0.6687            |  0.5194
> > SINE [2] | 0.5411         | 0.6512          |   0.5147
> > TGSRec [3] | 0.5372       | 0.6506           |   0.5121
> > LightSANs [4]  |0.5415 |0.6399 | 0.5119
> > RETR | 0.6103         | 0.7138          |   0.5822
> >
> >
> >
> > **Q6:** Replace the proposed pathway-based method with other sparse attention methods.
> >
> > As suggested by the reviewer, we replace the proposed pathway-based method with two sparse attention methods: LogSparse [5] and sparse attention [6].  We conduct experiments on Tmall:
> >
> > Method  | NDCG@10 ($\uparrow$)  | HR@10 ($\uparrow$) | MRR ($\uparrow$)
> > ---- | --- | --- | ---
> > RETR w/ LogSparse [5] | 0.4923        | 0.6015            |  0.4735
> > RETR w/ Sparse attention [6] | 0.4871        | 0.5873       |  0.4620
> > RETR | 0.6103         | 0.7138          |   0.5822
> >
> > The above results show that our RETR using the pathway attention remarkably outperforms other competing methods with two sparse attention methods. These sparse attention methods cannot capture the exact behavior pathway and show worse performance than RETR.
> >
> >
> >
> > **Q7:** Compare with graph-based methods.
> >
> > We add comparisons with the recent graph-based sequential recommendation models Jodie [7] and TGN [8] on all benchmarks. The full table is included revised paper.
> >
> > We here give snapshot results on Tmall (full comparisons can be found in $\underline{\textrm{Table 2}}$ in the revision). From these results, we can see that our RETR remarkably outperforms the SOTA graph-based methods.
> >
> >
> > Method  | NDCG@10 ($\uparrow$)  | HR@10 ($\uparrow$)  | MRR ($\uparrow$)
> > ---- | --- | --- | ---
> > Jodie [7] | 0.5307        | 0.6384            |  0.5003
> > TGN [8] | 0.5198         | 0.6362          |   0.4997
> > RETR | 0.6103         | 0.7138          |   0.5822
> >
> >
> > **Q8:**: More justifications to show that RETR will not be overwhelmed by trivial behaviors.
> >
> > We further provide visualization results for casual, correlated and drifted behavior pathways respectively in $\underline{\textrm{Figure 3 of the revised paper}}$. These three random samples from the steam dataset provide strong evidence that our RETR can capture various pathways and can avoid being overwhelmed by trivial behaviors.
> >
> > As suggested by the Reviewer sJpS, we evaluate our RETR using the captured behavior pathway as the inputs and get comparable results on the Tmall dataset below. This provides quantitative evidence that our RETR can effectively capture the various behavior pathway and not be overwhelmed by trivial behaviors.
> >
> > Method  | NDCG@10 ($\uparrow$) | HR@10 ($\uparrow$) | MRR ($\uparrow$)
> > ---- | --- | --- | ---
> > SASRec | 0.5049       |  0.6275          |  0.4804
> > RETR w/ pathway inputs |   0.6112     | 0.7142          |   0.5831
> > RETR | 0.6103 | 0.7138 | 0.5822

---

> > > ### Author Response · Authors · 2022-08-02
> > > **Our Response to Reviewer zis8**
> > >
> > > **Q9:** Results on large-scale datasets.
> > >
> > > As suggested by the reviewer, we conduct experiments on three large datasets: Netflix, MSD and Taobao.
> > > The detailed statistics of these datasets can be seen in Table 1 of the revised paper. Netflix, MSD and Taobao have 463,435, 571,355 and 987,994 users, which are obviously regarded as the large-scale dataset.
> > >
> > > We here give snapshot results on Netflix, MSD and Taobao (full comparisons can be found in $\underline{\textrm{Table 2 in the revision}}$ ). From these results, we can see that our RETR remarkably outperforms the SOTA recent recommendation methods.
> > >
> > >
> > > **Netflix**:
> > >
> > > Method  | NDCG@10 ($\uparrow$)  | HR@10 ($\uparrow$) | MRR ($\uparrow$)
> > > ---- | --- | --- | ---
> > > S3-Rec [1] | 0.3571       | 0.4917           |  0.2819
> > > SINE [2] | 0.3601        | 0.4902    |   0.2796
> > > TGSRec [3] | 0.3512       | 0.4887           |   0.2778
> > > LightSANs [4]  |0.3441 |0.4852 | 0.2785
> > > RETR | 0.3725         | 0.5142         |   0.3134
> > >
> > >
> > > **MSD**:
> > >
> > > Method  | NDCG@10 ($\uparrow$)  | HR@10 ($\uparrow$) | MRR ($\uparrow$)
> > > ---- | --- | --- | ---
> > > S3-Rec [1] | 0.5381       | 0.5315           |  0.3494
> > > SINE [2] | 0.5304        | 0.5264    |   0.3667
> > > TGSRec [3] | 0.5279       | 0.5137          |   0.3612
> > > LightSANs [4]  |0.5163 |0.4994 | 0.3451
> > > RETR | 0.5981        | 0.5912        |   0.3901
> > >
> > >
> > > **Taobao**:
> > >
> > > Method  | NDCG@10 ($\uparrow$)  | HR@10 ($\uparrow$) | MRR ($\uparrow$)
> > > ---- | --- | --- | ---
> > > S3-Rec [1] | 0.0827       | 0.1336           |  0.0919
> > > SINE [2] | 0.0873        | 0.1580    |   0.0934
> > > TGSRec [3] | 0.0745       | 0.1537           |   0.0802
> > > LightSANs [4]  |0.0694 |0.1590 | 0.0741
> > > RETR | 0.1195       | 0.1768       |   0.1117
> > >
> > >
> > > **Q10:** Minor issues.
> > >
> > > Thanks for your valuable suggestions. We have fixed these issues in the revised paper and added the ablation study of multi-head attention in Appendix A.
> > >
> > > ---
> > >
> > > [1] Kun Zhou, Hui Wang, Wayne Xin Zhao, Yutao Zhu, Sirui Wang, Fuzheng Zhang, Zhongyuan Wang, Ji-Rong Wen, “S3-Rec: Self-Supervised Learning for Sequential Recommendation with Mutual Information Maximization,” CIKM 2020
> > >
> > > [2] Qiaoyu Tan, Jianwei Zhang, Jiangchao Yao, Ninghao Liu, Jingren Zhou, Hongxia Yang, Xia Hu, “Sparse-Interest Network for Sequential Recommendation,” WSDM 2021
> > >
> > > [3] Ziwei Fan, Zhiwei Liu, Jiawei Zhang, Yun Xiong, Lei Zheng, Philip S. Yu, “Continuous-Time Sequential Recommendation with Temporal Graph Collaborative Transformer,” CIKM 2021
> > >
> > > [4] Xinyan Fan, Zheng Liu, Jianxun Lian, Wayne Xin Zhao, Xing Xie, and Ji-Rong Wen, "Lighter and better: low-rank decomposed self-attention networks for next-item recommendation," SIGIR 2021
> > >
> > > [5] Li, Shiyang, et al. "Enhancing the locality and breaking the memory bottleneck of transformer on time series forecasting." NeurIPS 2019.
> > >
> > > [6]Child, Rewon, et al. "Generating long sequences with sparse transformers." arXiv preprint arXiv:1904.10509 (2019).
> > >
> > > [7] Kumar, Srijan, Xikun Zhang, and Jure Leskovec. "Predicting dynamic embedding trajectory in temporal interaction networks." Proceedings of the 25th ACM SIGKDD international conference on knowledge discovery & data mining. 2019.
> > >
> > > [8] Emanuele Rossi, Ben Chamberlain, Fabrizio Frasca, Davide Eynard, Federico Monti, and Michael Bronstein. Temporal graph networks for deep learning on dynamic graphs. arXiv preprint arXiv:2006.10637,412 2020

---

### Official Review · Reviewer_uSox · 2022-07-07

**Rating:** 7
**Confidence:** 3
**Soundness:** 3 good
**Presentation:** 3 good
**Contribution:** 3 good

**Summary:**

This paper proposes a recommender transformer with a pathway attention mechanism. It is characterized by its ability to capture three types of user pathways and predict user action sequences with high accuracy. The paper demonstrates the usefulness of the proposed method in comparison with several state-of-the-art methods using several types of real data.

**Questions:**

The authors state that user pathways fall into three categories (correlated, casual, and drifted behavior pathways), but on what evidence? Justification is needed.

**Limitations:**

There is no mention in the paper of negative impacts on society. Also, I can't think of any.

**Strengths And Weaknesses:**

- Strengths
  - Starting with the actual example in Figure 1, the motivation for proposing the method is well explained, making it easy to understand the content of the proposed technique.
  - The authors have conducted prediction experiments using seven different behavioral log datasets from various sites. They compared the accuracy of the proposed method with seven existing methods and confirmed that the proposed method outperforms them.
  - Experiments are conducted using real data, not artificial data.
- Weaknesses
  - The meaning and boundaries of the three types of pathways are vague. The definition of each should be clearly stated. Also, are these three types sufficient?
For example, what is the difference between the Correlated behavior pathway and the Drifted behavior pathway? They seem to have similar properties in the local and short-term. The types and definitions of pathways should be better justified, such as providing references that support the authors' definitions.
  - Although the experimental results quantitatively demonstrate the effectiveness of the proposed method, the architecture in Figure 2 is straightforward and somewhat lacking in technical novelty.

---

> ### Author Response · Authors · 2022-08-02
> **Our Reponse to Reviewer uSox**
>
> **Q1:** The authors state that user pathways fall into three categories (correlated, casual, and drifted behavior pathways), but on what evidence? Justification is needed.
>
> We use these three categories to covering the representative types of user behaviors.
>
> - A user may be randomly or regularly interested in a particular item.
> For the random interests, we define this phenomenon as the Casual Behavior Pathway.
> The random interests lead to the casual behavior pathway.
>
> - If a user is regularly interested in a particular item, the user will be interested in it for a certain period. We define this phenomenon as the Correlated Behavior Pathway.
>
> - Otherwise, the user's interest is evolving over time, which is widely found in previous recommendation methods like SMRec [1]. A user’s behaviors in a particular period might drift over time and the user will be interested in another item. We conclude this phenomenon as the Drifted Behavior Pathway.
>
> We further showcase these three types of user behaviors from datasets in $\underline{\textrm{Figure 3 of the revised paper}}$.
>
>
>
> **Q2:** The types and definitions of pathways should be better justified.
>
> We give the **detail defination** for each behavior pathway:
>
> - **Casual behavior pathway**: The user clicks a particular class of items randomly at casual times. These clicked behaviors are not clicked continuously.
>
> - **Correlated behavior pathway**: The user clicks a particular class of items continuously for a certain period.
>
> - **Drifted behavior pathway**: The user clicks a particular class of items continuously at a certain period. After that time, the user starts to click another particular class of items continuously for a certain period.
>
> **Q3:** What is the difference between the Correlated behavior pathway and the Drifted behavior pathway?
>
> The drifted behavior pathway is different from the correlated behavior pathway because it considers the evolving interests in the long-range behaviors. On the contrary, the correlated behavior pathway mainly focuses on the stable interests in the short-range behaviors.
>
>
> ---
>
> [1] Chao Chen, Haoyu Geng, Nianzu Yang, Junchi Yan, Daiyue Xue, Jianping Yu, and Xiaokang Yang. Learning self-modulating attention in continuous time space with applications to sequential recommendation. ICML 2021

---

### Official Review · Reviewer_bU2p · 2022-07-11

**Rating:** 3
**Confidence:** 5
**Soundness:** 2 fair
**Presentation:** 3 good
**Contribution:** 2 fair

**Summary:**

In this paper, the authors propose the Recommender Transformer (RETR) with a novel Pathway Attention mechanism. RETR can dynamically plan the behavior pathway specified for each user, and sparingly activate the network through this behavior pathway to effectively capture evolving patterns useful for recommendation.

**Questions:**

 It is unclear why need use this swith rounter in sequential recomemndation. Usually, the length of users' behaviors is very short, like less than 25 for most of transaction. Do we really need this router in the Transformer?

Actually, recent work verify that a simle MLP can outperform the Transformer in the sequential recommendation.

Zhou, Kun, et al. "Filter-enhanced MLP is all you need for sequential recommendation." Proceedings of the ACM Web Conference 2022. 2022.


3. In the experiment, "we pair the ground-truth item with 100 randomly sampled negative items that the user has not interacted with." Does this raise a smapling bias?

**Ethics Review Area:**

["I don’t know"]

**Limitations:**

Yes

**Strengths And Weaknesses:**

Strength:

1. The paper is well written and easy to follow. Basically, the authors try to use the behavior pathway in the Transformer.

2. The experimental results are good compared with the baselines.


Weakness:
1. The novelty of this work is not very high. The mechanism of router is wildey used in the MoE-style Transformer. It seems that this work applied it to the sequential recommendation. The Gumbel-softmax is also widely used to optimize the discrete binary variable.

2. It is unclear why need use this swith rounter in sequential recomemndation. Usually, the length of users' behaviors is very short, like less than 25 for most of transaction. Do we really need this router in the Transformer?

Actually, recent work verify that a simle MLP can outperform the Transformer in the sequential recommendation.

Zhou, Kun, et al. "Filter-enhanced MLP is all you need for sequential recommendation." Proceedings of the ACM Web Conference 2022. 2022.

The motivation of this work is thus not very strong.

3. In the experiment, "we pair the ground-truth item with 100 randomly sampled negative items that the user has not interacted with." Does this raise a smapling bias?

---

> ### Author Response · Authors · 2022-08-02
> **Our Response to Reviewer bU2p**
>
> **Q1:** It is unclear why we need to use this swith rounter in sequential recommendation.
>
> Previous sequential recommendation methods have proved that the recommender can be benefited a lot from the user’s historical behaviors, even though the behavior sequence may be short. However, when meeting with the short behavior sequence, the recommender still needs to deal with various behavior pathways and can be overwhelmed by the trivial behaviors.
>
> As shown in $\underline{\textrm{Appendix Figure 1}}$ in the revised paper, we show the last 10 behaviors from a random user in the Steam dataset. Further, in $\underline{\textrm{Appendix Figure 1}}$  we show that the state-of-the-art MLP-based model, FMLP-Rec [1], is still overwhelmed by the old drifted behaviors (simulation games).
>
> To avoid the recommender being overwhelmed by the trivial behaviors, we design the pathway router to capture the pivotal behavior pathway that explains the user's preferences, whenever the behavior sequence is short or long. It's crucial to develop the pathway router to capture the behavior pathway for making precise recommendations.
>
> Our pathway router is a **general module** which is designed not only for the Transformers. It can also enhance MLP-based model. We apply the pathway router to the state-of-the-art MLP-based model, FMLP-Rec [1], and evaluate its performance on Taobao, MovieLens1M and Yelp.
>
>
>
> Taobao:
>
> Method  | NDCG@10 (higher is better  | HR@10 (higher is better) | MRR (higher is better)
> ---- | --- | --- | ---
> FMLP-Rec[1] | 0.0678        | 0.1421         |  0.0603
> FMLP-Rec + pathway router | 0.0893         | 0.1659        |   0.0834
> RETR | 0.1195         | 0.1768          |   0.1117
>
> MovieLens1M
>
> Method  | NDCG@10 (higher is better  | HR@10 (higher is better) | MRR (higher is better)
> ---- | --- | --- | ---
> FMLP-Rec[1] | 0.5948        | 0.6043           |  0.5519
> FMLP-Rec + pathway router | 0.6217         | 0.8293          |   0.5704
> RETR | 0.6351         | 0.8467          |   0.5921
>
> Yelp:
>
> Method  | NDCG@10 (higher is better  | HR@10 (higher is better) | MRR (higher is better)
> ---- | --- | --- | ---
> FMLP-Rec[1] | 0.5024        | 0.7720           |  0.4299
> FMLP-Rec + pathway router | 0.5225         | 0.7946          |   0.4502
> RETR | 0.5136         | 0.7730          |   0.4354
>
>
> From the table above, we can see that FMLP-Rec can benefit a lot from our pathway router. The pathway router is important for sequential recommendation models to avoid being overwhelmed by trivial user behaviors.
>
>
>
> **Q2:** In the experiment, "we pair the ground-truth item with 100 randomly sampled negative items that the user has not interacted with." Does this raise a sampling bias?
>
> In the previous literature, this sampling strategy is widely used. To avoid heavy computation on all user-item pairs, we followed the strategy used in SASRec. For each user, we randomly sample 100 negative items, and rank these items with the ground-truth item. According to the rankings of these 101 items, HR@10 and NDCG@10 can be evaluated. This is a *de facto* configuration for sequential recommendation.
>
> For fairness, we adopt the same sampling strategy for all comparing models to evaluate the performance. All in all, this sampling strategy will not raise a sampling bias.
>
>
>
> **Q3:** Clarify the novelty of the architecture.
>
> - Previous MoE-style Transformers like Switch Transformer [3] usually adjust the pathway towards different feedforward networks (FFNs). On the contrary, our RETR routes the query for the pathway attention instead of routing the FFNs. Note that RETR does not route the network but routes the exact user behaviors. Furthermore, the behavior pathway is routed hierarchically towards the feedforward procedure, while this hierarchical procedure is not used in previous MoE-style Transformers.
>
> - Previous methods like [2] often use the Gumble-Softmax to decide which task to choose. Our RETR is the first to route the user's behavior pathway using the pathway router. In contrast, the Gumble-Softmax is adopted in our work to decide whether each behavior can be selected as the behavior pathway. Since reasoning about the exact behavior pathway is crucial to the sequential recommendation performance, our use of the Gumbel-Softmax strategy to capture the behavior pathway brings new ideas and practical guides to the recommendation literature.
>
>
> ---
> [1] Zhou, Kun, et al. "Filter-enhanced MLP is all you need for sequential recommendation." Proceedings of the ACM Web Conference, 2022.
>
> [2] Shen, Jiayi, et al. "Variational multi-task learning with gumbel-softmax priors." NeurIPS, 2021.
>
> [3]Fedus, William, Barret Zoph, and Noam Shazeer. "Switch transformers: Scaling to trillion parameter models with simple and efficient sparsity." JMLR, 2021.

---

> ### Author Response · Authors · 2022-08-08
> **Kind request for reviewer feedback to our rebuttal**
>
> Dear Reviewer,
>
> Many thanks for your time and efforts in reviewing our paper. Your reviews are highly instructive to improve our paper greatly.
>
> We kindly remind that we are at the **final stage of discussion (Aug 3rd-9th)** and have only a few days for the discussion. We have made an exhaustive effort to try to successfully address your concerns and answer your questions, by providing all supporting experiments you requested and clarifying all questions you asked. **We have proved that the recommender still needs to deal with various behavior pathways and can be overwhelmed by trivial behaviors, whenever the behavior sequence is short or long. The MLP-based model like FMLP-Rec can also benefit from our pathway module.**
>
> If you have any further concerns or questions, please do not hesitate to let us know, and we will be happy to answer them timely.
>
> All the best,
> Authors

---

> ### Author Response · Authors · 2022-08-09
> **Discussion period ending soon**
>
> Dear Reviewer,
>
> Thank you once again for your review of our work. As the discussion period is approaching its end, we would be grateful if you could confirm whether our responses and the additions we have made to the manuscript addressed your concerns, and let us know if any issues remain.
>
> Thanks again for your time and reviews.

---

> ### Author Response · Authors · 2022-08-10
> **Looking forward to further comments**
>
> Dear Reviewer,
>
> We are sincerely looking forward to your efforts in reviewing our paper. We have provided corresponding responses and results, which we believe have covered your concerns. We hope to have a further discussion with you about whether your concerns have been clarified or not. Please let us know if you still have any unclear issues with our work.
>
>
> Thanks again for your time and reviews.

---

### Official Review · Reviewer_sJpS · 2022-07-12

**Rating:** 5
**Confidence:** 4
**Soundness:** 2 fair
**Presentation:** 3 good
**Contribution:** 2 fair

**Summary:**

This paper proposes a new sequential recommendation model with behavior pathways, effectively capturing specific evolving item patterns for each user. The pathway attention using learned binary routes can effectively remove unnecessary items for a given user sequence. Experimental results show that the proposed model significantly outperforms existing models on seven benchmark datasets and achieves state-of-the-art performance. Overall, the idea of using behavior pathways is interesting to me, but it has several weak points, especially in evaluating the proposed model.

**Questions:**

- Q1) Is V^t in Eq. (4) the same as hat(R^l) in Eq. (5)? If so, please clarify this equation.
- Q2) The proposed idea seems to be model-agnostic. Could you apply the pathway module for other sequential recommender models, e.g., BERT4Rec?
- Q3) In Figure 1, the authors mention various behavior pathways. Does the proposed model capture various pathways or focus on capturing drifted behavior pathways?


**Limitations:**

This paper does not address the negative societal impact. However, this paper seems not to have any negative impact.

**Strengths And Weaknesses:**

Strengths
- (Originality) It is interesting to introduce behavior pathways in long user sequences. Besides, this paper utilizes a router to choose item patterns selectively.
- (Originality) Because the idea of using the router is a model-agnostic property, it can be effectively applied to various sequential recommender models.
- (Clarity) It is well-written and easy to understand.


Weaknesses
- (Quality) Although the evaluation is extensive, the proposed model has not been compared with recent sequential recommendation models. Please refer to the following references.
[1] Kun Zhou, Hui Wang, Wayne Xin Zhao, Yutao Zhu, Sirui Wang, Fuzheng Zhang, Zhongyuan Wang, Ji-Rong Wen, “S3-Rec: Self-Supervised Learning for Sequential Recommendation with Mutual Information Maximization,” CIKM 2020
[2] Qiaoyu Tan, Jianwei Zhang, Jiangchao Yao, Ninghao Liu, Jingren Zhou, Hongxia Yang, Xia Hu, “Sparse-Interest Network for Sequential Recommendation,” WSDM 2021
[3] Ziwei Fan, Zhiwei Liu, Jiawei Zhang, Yun Xiong, Lei Zheng, Philip S. Yu, “Continuous-Time Sequential Recommendation with Temporal Graph Collaborative Transformer,” CIKM 2021
- (Quality) In Section 4.3, the visualization result shows that the proposed model effectively captures the pathways in user sequences. However, it is wondering if this result is generalized for other cases. It is necessary to show a quantitative result on whether the proposed model effectively captures a useful pathway. One possible evaluation is that the proposed model also shows a comparable result using a subset of sequences derived from a pathway.
- (Quality) In Table 3, RETR(L=1) is similar to SASREC(L=1). However, RETR(L=1) shows a better performance than SASRec(L=1). It is wondering if the proposed model is fairly compared with the existing model.
- (Clarify) There are some minor typos.
- 1p 37line: use’s -> user’s
- 2p 41line: second -> first
- 2p 44line: first -> second

---

> ### Author Response · Authors · 2022-08-02
> **Our Response to Reviewer sJpS**
>
> **Q1:** Is $v_t$ in Eq. (4) the same as $\widehat{\mathcal{R}}_{t}^{l}$ in Eq. (5)? If so, please clarify this equation.
>
> No, $v_t$ in Eq. (4) is not the same as $\widehat{\mathcal{R}}_{t}^{l}$ in Eq. (5).
>
> As described in the main context of Eq. (3), the $\widehat{\mathcal{R}}_{t}^{l}$ is obtained by the argmax operation during the feedforward procedure.
>
> However, the argmax operation is non-differentiable. To make the backward procedure of the Gumbel-Softmax differentiable, the Gumbel-Softmax calculates the gradient from Eq. (4), which is a differentiable approximation to relax $\widehat{\mathcal{R}}_{t}^{l}$ to $v_t$. The whole procedure can be regarded as the reparameterization trick widely used in deep learning.
>
>
> **Q2:** The proposed idea seems to be model-agnostic. Could you apply the pathway module for other sequential recommender models, e.g., BERT4Rec?
>
> As suggested by the reviewer, we apply the pathway module for BERT4Rec and S3-Rec on MovieLens
>
> Method  | NDCG@10 ($\uparrow$)  | HR@10 ($\uparrow$) | MRR ($\uparrow$)
> ---- | --- | --- | ---
> BertRec | 0.5965        | 0.8269            |  0.5614
> BertRec + Pathway | 0.6376         | 0.8491          |   0.5972
> S3-Rec[1] | 0.6103        | 0.8312           |   0.5729
> S3-Rec[1] + Pathway | 0.6482         | 0.8577           |   0.6048
>
> In the above table, we observe that our pathway module can improve the performance of BertRec and S3-Rec substantially. RETR can be further enhanced using advanced backbones alternative to the vanilla Transformers.
>
>
> **Q3:** In Figure 1, the authors mention various behavior pathways. Does the proposed model capture various pathways or focus on capturing drifted behavior pathways?
>
> Our RETR can capture various pathways. We further provide the qualitative results for casual, correlated, and drifted behavior pathways respectively in $\underline{\textrm{Figure 3 of revised paper}}$:
>
> - **Casual behavior pathway**: An example of the casual behavior pathway is shown in Figure 3(a). The RGB game is randomly clicked at casual times. Our RETR can capture all the RPG behavior pathway, while the SASRec focuses on the wrong recent adventure games. The SASRec cannot capture the early clicked RPG game.
>
> - **Correlated behavior pathway**: An example for correlated behavior pathway is shown in Figure 3(b). The indie game is clicked many times recently, leading to the final decision to an indie game. Our RETR can effectively capture the correlated behavior pathway. However, the SASRec provides higher attention scores on the recent RPG and games. On the contrary, our RETR pay no attention on these wrong results.
>
> - **Drifed behavior pathway**: An example for drifted behavior pathway is shown in Figure 3(c). The user was initially interested in the indie game, but suddenly became interested in simulation games recently and chose an indie game at last. Our RETR captures the drifted behavior pathway for the indie game and has not concentrated on the old drifted pathway -- simulation games. However, the SASRec gives more attention to the simulation games, making it being overwhelmed by trival user behaviors.
>
> The above three random examples from the steam dataset provides the strong evidence that our RETR can capture various pathways.
>
>
> **Q4:** Compare with recent sequential recommendation models.
>
> We add comparisons with the recent sequential recommendation models S3-Rec, SINE, TGSRec on all benchmarks. The full table is included in the $\underline{\textrm{revised paper}}$.
>
> We here give snapshot results on Yelp, MovieLens1M, and Tmall (full comparisons can be found in $\underline{\textrm{Table 2}}$ in the revision). It can be observed that our RETR remarkably outperforms the SOTA methods.
>
>
> Yelp:
>
> Method  | NDCG@10 ($\uparrow$)  | HR@10 ($\uparrow$) | MRR ($\uparrow$)
> ---- | --- | --- | ---
> S3-Rec [1] | 0.4937        | 0.7597            |  0.4107
> SINE [2] | 0.4902         | 0.7564          |   0.4093
> TGSRec [3] | 0.4887       | 0.7533           |   0.4072
> RETR | 0.5136         | 0.7730          |   0.4354
>
>
> MovieLens1M:
>
> Method  | NDCG@10 ($\uparrow$)  | HR@10 ($\uparrow$) | MRR ($\uparrow$)
> ---- | --- | --- | ---
> S3-Rec [1] | 0.6172        | 0.8352            |  0.5812
> SINE [2] | 0.6134         | 0.8311         |   0.5801
> TGSRec [3] | 0.6081       | 0.8303           |   0.5734
> RETR | 0.6351         | 0.8467          |   0.5921
>
>
>
> Tmall:
>
> Method  | NDCG@10 ($\uparrow$)  | HR@10 ($\uparrow$) | MRR ($\uparrow$)
> ---- | --- | --- | ---
> S3-Rec [1] | 0.5423        | 0.6687            |  0.5194
> SINE [2] | 0.5411         | 0.6512          |   0.5147
> TGSRec [3] | 0.5372       | 0.6506           |   0.5121
> RETR | 0.6103         | 0.7138          |   0.5822

---

> > ### Author Response · Authors · 2022-08-02
> > **Our Response to Reviewer sJpS**
> >
> > **Q5:** Quantitative results on whether the proposed model effectively captures a useful pathway.
> >
> > As suggested by the reviewer, we give quantitive results to validate that our RETR can effectively capture various behavior pathways. We evaluate our RETR using a subset of sequences derived from the obtained behavior pathway on Tmall.
> >
> > Technically, we first train RETR on Tmall. For each user, we take the captured behavior pathway from our RETR as the inputs to retrain a RETR rather than using the whole user's behaviors. From the results below, we find that using the behavior pathway as the inputs can achieve comparable results as the original RETR which uses complete user behaviors. It provides the evidence that our RETR can aptly capture the useful pathway for each user.
> >
> > Method  | NDCG@10 ($\uparrow$)  | HR@10 ($\uparrow$) | MRR ($\uparrow$)
> > ---- | --- | --- | ---
> > SASRec | 0.5049       |  0.6275          |  0.4804
> > RETR w/ pathway inputs |   0.6112     | 0.7142          |   0.5831
> > RETR | 0.6103 | 0.7138 | 0.5822
> >
> >
> >
> > **Q6:** If the proposed model is fairly compared with the existing model.
> >
> > Our RETR (L=1) is different from SASRec (L=1). Each RETR block has a pathway router to route the query from the original inputs. The pathway attention of RETR is cross-attention between the pathway tokens and off-pathway tokens, while SASRec uses the self-attention without the pathway router.
> >
> > In $\underline{\textrm{Table 3}}$, SASRec achieves the best performance when L=2 on Yelp. For fairness, the hyperparameter for all comparing models is carefully tuned to achieve their best performance. The proposed model is fairly compared with the existing model.
> >
> > Method  | MRR ($\uparrow$)
> > ---- | ---
> > SASRec (L=1) |  0.3813
> > SASRec (L=2) |  0.3927
> > SASRec (L=3) |  0.3922
> > SASRec (L=4) |  0.3919
> >
> >
> > **Q7:** Minor typos.
> >
> > We have carefully fixed these typos in the revised paper.
> >
> >
> > ---
> > [1] Kun Zhou, Hui Wang, Wayne Xin Zhao, Yutao Zhu, Sirui Wang, Fuzheng Zhang, Zhongyuan Wang, Ji-Rong Wen, “S3-Rec: Self-Supervised Learning for Sequential Recommendation with Mutual Information Maximization,” CIKM 2020
> >
> > [2] Qiaoyu Tan, Jianwei Zhang, Jiangchao Yao, Ninghao Liu, Jingren Zhou, Hongxia Yang, Xia Hu, “Sparse-Interest Network for Sequential Recommendation,” WSDM 2021
> >
> > [3] Ziwei Fan, Zhiwei Liu, Jiawei Zhang, Yun Xiong, Lei Zheng, Philip S. Yu, “Continuous-Time Sequential Recommendation with Temporal Graph Collaborative Transformer,” CIKM 2021

---

> ### Author Response · Authors · 2022-08-07
> **Discussion period ending soon**
>
> Dear Reviewer,
>
> Thanks again for the time and reviews. Since the final stage of discussion is ending soon, please let us know if our response has addressed your concerns.
>
> As requested, we have made every effort to prove that our RETR has the ability to capture various behavior pathways effectively. With more supporting experiments, we showed that our RETR could be further enhanced using an advanced backbones alternative to the vanilla Transformers.
>
> Besides, we provide the quantitive results to validate that our RETR can effectively capture various behavior pathways following your request. The more supporting qualitative visualization results are also provided in $\underline{\textrm{Figure 3 of the revised paper}}$. We further included comparisons with S3-Rec, SINE, and TGSRec on all benchmarks.
>
>
> We will be happy to answer if there are additional issues/questions.

---

> > ### Author Response · Authors · 2022-08-08
> > **Kind request for reviewer feedback to our rebuttal**
> >
> > Dear Reviewer,
> >
> > The final stage of discussion is ending soon, so please kindly let us know if our response has addressed your concerns. We will be happy to answer if there are additional issues/questions.
> >
> >
> >
> > Thanks again for your time and reviews.

---

> > > ### Comment · Reviewer_sJpS · 2022-08-09
> > > **Additional questions**
> > >
> > > It is very grateful for the detailed answers to each question. Most of the questions have been resolved. Therefore, I raised the score.
> > >
> > > Additionally, I still have some questions.
> > >
> > > Q1. Could you clarify the difference between RETR(L=1) and SASRec(L=1)?
> > >
> > > I understand that the different part between SASRec and RETR is that RETR uses pathway attention instead of self-attention. Specifically, the difference is the query, while the key and value are the same in both attention methods.
> > >
> > > In particular, when L=1, for the key and value, the two methods are exactly the same. (since they use raw behavior embedding Z^0=X_s)
> > >
> > > Both use Z_t^1 as the final user representation. Z_t^1 is the same in the two methods if RETR processes the last item (=t-th item) on-pathway, but it is different if the last item is processed off-pathway.
> > >
> > > Q2. For table 2, how many seeds have you conducted?

---

> > > > ### Author Response · Authors · 2022-08-09
> > > > **Our Response to Reviewer sJpS**
> > > >
> > > > Thanks again for your kind suggestions and insightful comments, which have greatly inspired us to improve our work.
> > > >
> > > > Q1. Further clarify the difference between RETR (  $L=1$ ) and SASRec ( $L=1$ ).
> > > >
> > > > We clarify that the difference between the pathway attention in RETR ( $L=1$ ) and self-attention in SASRec ( $L=1$ ) is the query, while the key and value are the same in both attention methods. We adaptively route the query tokens for the pathway attention with the behavior pathway to make the RETR concentrate more on the behavior pathway rather than trivial behaviors.
> > > >
> > > > However, the last item prediction is also influenced by other prediction pairs.  Technically, we train both the RETR ( $L=1$ ) and SASRec ( $L=1$ ) using the pairwise ranking loss in Eq. (7). These predictive models are trained with different prediction pairs $1 \rightarrow 2$, $ 2 \rightarrow 3$, ... , $N-1 \rightarrow N$. Here $t-1 \rightarrow t$ means predicting the $t$-th item only conditioned on the first $t-1$ items.
> > > >
> > > > Thus, the training process of the representation $\mathcal{Z}_N$ of the last item ( $N$ is the length) is also influenced by the training process of different $\mathcal{Z}_t$, $t=1,2,3,..,N-1$. (The superscript $L=1$ omitted for clarity).
> > > > **By incorporating the pathway module, our pathway attention is trained differently compared to the self-attention using different prediction pairs, leading to different weight parameters from the RETR ( $L=1$ ) and SASRec ( $L=1$ ).** The $\mathcal{Z}_N$ in the RETR ( $L=1$ ) will be different from $\mathcal{Z}_N$ in the SASRec ( $L=1$ ) with different weight parameters.
> > > >
> > > >
> > > > Q2: For Table 2, how many seeds have you conducted?
> > > >
> > > > For all baselines and RETR in Table 2, we have conducted three random seeds and reported the average results.

---

> > > > ### Author Response · Authors · 2022-08-09
> > > > **Thanks for your reviews**
> > > >
> > > > Dear Reviewer
> > > >
> > > > Your academic suggestions inspire us a lot and make our paper strong. We appreciate your great efforts and really enjoy having a discussion with you.
> > > >
> > > >
> > > > Thanks again for your time and reviews

---

### Author Response · Authors · 2022-08-02
**Summary of Changes**

We thank all reviewers for their constructive comments. Accordingly, we have revised the paper substantially and updated the new version. Please check out the following changes  (highlighted in green color) in the revised paper:

1. We compare RETR with more recent SOTA sequential recommendation models, including
S3-Rec, SINE, TGSRec, Jodie, TGN. Quantitative results are shown in Table 2 and detailed analyses are shown in $\underline{\textrm{Section 4.1}}$.

2. We add quantitative experimental results on large-scale real-world datasets (Netflix, MSD and Taobao), and each dataset contains a large number of users. Results can be found in $\underline{\textrm{Table 2}}$.

3. We provide three typical examples corresponding to the casual, correlated and drifted behavior pathways respectively in $\underline{\textrm{Figure 3}}$. The detail explanations are in $\underline{\textrm{Section 4.3}}$. We also add visualizations in $\underline{\textrm{Figure 1 of Appendix}}$, which shows that the recent MLP-based method FMLP-Rec is still overwhelmed by trivial behaviors.

4. We provide the ablation study for multi-head attention in $\underline{\textrm{Appendix A}}$.

5. We moved some contents in the original paper to the appendix in the revision: the experiment results on the Beauty, Sports and Toys dataset (in $\underline{\textrm{Appendix B}}$ now).


If there are any additional comments on the revision, please do not hesitate to let us know. We are glad to answer any further questions.

---

### Meta-Review · Area_Chair_pUU6 · 2022-08-26

**Recommendation:** Reject
**Confidence:** Certain

**Metareview:**

This paper presents Recommender Transformer (RETR) with a pathway attention mechanism that can dynamically zeroing-out the interactions (e.g., the trivial/noisy ones) in transformer-based sequential recommender systems. Extensive experimental results demonstrate the effectiveness of the proposed architecture.

Overall this paper received mixed reviews with borderline scores. The reviewers raised concerns around baselines and evaluations, some of which the authors promptly addressed in the revision during the rebuttal period. I also read the paper in details myself. I do agree with some of the concerns from the reviewers but I don't think a method needs to beat every other published papers to be published (and I think the current baselines are more than thorough enough). My biggest complaint about the paper is around the writing, specifically, how the proposed idea is presented.

This paper tries to tackle an important question, which is that in sequential recommendation, not every interactions are useful in helping predict future interaction. The self-attention mechanism in transformer kind of addresses this problem but in a more "softer" fashion with attention weights. This paper presents a simple yet effective method to introduce a pathway mechanism that adaptively zeroing-out some of the interactions via a binary pathway router. In order to train such a model end-to-end, Gumbel-softmax sampling is utilized.

The most important part of the contribution to me is that this is an improvement to the transformer architecture, as opposed to a new model which is what this paper's writing suggests -- the proposed approach is effectively model-agonistic and doesn't marry to a particular loss function or finer-grained architectural choices (number of layers, etc.). Currently there are many baselines in the paper, but each made some different model/architecture choices, which could contribute to the difference in performance (or not, but we wouldn't know). An ideal evaluation should have been to take all the transformer-based baselines that are currently in the paper, add this pathway mechanism without changing anything else, and show that the results improved over the transformer architecture. In this way, we know the improvements are exactly coming from introducing the pathway. The authors might argue some of the current results are already supporting this argument, but my point is to emphasize this point very explicitly rather than leaving it for the readers to infer.

From what I read in this paper, I truly believe this pathway idea has its potential. Therefore, I would especially want the authors to further refine the presentation to better convey the idea, which in turn will hopefully increase the impact of this paper once it is eventually published.

Some minor comments:
* The way the paper is currently written seems to suggest there are only three types of pathways and the network is capable of capturing all of them. I am personally not a big of fan of over-interpreting what a neural net is trying to do. Therefore, I wouldn't overly focus on the characterization of different pathways and only show the qualitative examples at the end as a high-level demonstration.
* In Eq 2 "softmax" should really be "sigmoid" if a 0-1 prediction is made there. Then the following line "logit" is probably not the right word here.
* The qualitative examples at the end (figure 3) can be more carefully examined/labeled. For example, the current categorization is quite ambiguous -- "Indie" refers to the type of developers while "JPG" refers to the genre of the game, they are certainly not mutually exclusive.

**Award:**

No

---

### Decision · Program_Chairs · 2022-09-14

Reject